# Microfluidics-Assisted Size Tuning and Biological Evaluation of PLGA Particles

**DOI:** 10.3390/pharmaceutics11110590

**Published:** 2019-11-08

**Authors:** Maria Camilla Operti, Yusuf Dölen, Jibbe Keulen, Eric A. W. van Dinther, Carl G. Figdor, Oya Tagit

**Affiliations:** 1Department of Tumor Immunology, Radboud Institute for Molecular Life Sciences, Radboud University Medical Center, 6500 HB Nijmegen, The Netherlands; MariaCamilla.Operti@radboudumc.nl (M.C.O.); Yusuf.Dolen@radboudumc.nl (Y.D.); jibbekeulen@gmail.com (J.K.); Eric.vanDinther@radboudumc.nl (E.A.W.v.D.); Carl.Figdor@radboudumc.nl (C.G.F.); 2Oncode Institute, 3553 Utrecht, The Netherlands

**Keywords:** PLGA, drug delivery systems, microfluidics, nanoparticles, microparticles

## Abstract

Polymeric particles made up of biodegradable and biocompatible polymers such as poly(lactic-co-glycolic acid) (PLGA) are promising tools for several biomedical applications including drug delivery. Particular emphasis is placed on the size and surface functionality of these systems as they are regarded as the main protagonists in dictating the particle behavior in vitro and in vivo. Current methods of manufacturing polymeric drug carriers offer a wide range of achievable particle sizes, however, they are unlikely to accurately control the size while maintaining the same production method and particle uniformity, as well as final production yield. Microfluidics technology has emerged as an efficient tool to manufacture particles in a highly controllable manner. Here, we report on tuning the size of PLGA particles at diameters ranging from sub-micron to microns using a single microfluidics device, and demonstrate how particle size influences the release characteristics, cellular uptake and in vivo clearance of these particles. Highly controlled production of PLGA particles with ~100 nm, ~200 nm, and >1000 nm diameter is achieved through modification of flow and formulation parameters. Efficiency of particle uptake by dendritic cells and myeloid-derived suppressor cells isolated from mice is strongly correlated with particle size and is most efficient for ~100 nm particles. Particles systemically administered to mice mainly accumulate in liver and ~100 nm particles are cleared slower. Our study shows the direct relation between particle size varied through microfluidics and the pharmacokinetics behavior of particles, which provides a further step towards the establishment of a customizable production process to generate tailor-made nanomedicines.

## 1. Introduction

As evidenced by the dramatic increase in the number of studies and clinical applications over time, polymeric particles play a crucial role in drug delivery, providing improved stability, targeted delivery, and sustained release of loaded therapeutic agents without causing off-target toxicities in vivo [1,2,3,4,5]. Drug delivery particles based on poly(lactic-co-glycolic acid) (PLGA) are among the most commonly studied vehicles due to excellent biocompatibility, tuneable degradation characteristics, and long clinical history of PLGA [6,7,8]. The remarkable physicochemical properties and high versatility of PLGA have proven to address several challenges in drug delivery. PLGA can be processed into almost any size and shape [7] and can encapsulate molecules of virtually any size such as small drugs [9,10,11,12,13], proteins [14,15,16,17,18], nucleic acids [19,20,21,22,23], and vaccines [24,25,26,27,28,29]. The colloidal features such as size and surface functionality have a direct influence on the cellular uptake, biodistribution, and thus therapeutic efficacy of the particles, dictating the in vivo fate of therapeutic cargo [30,31]. These colloidal features are particularly important in cancer immunotherapy as the efficacy of any given therapeutic agent is highly dependent on its ability to reach either the tumor microenvironment (TME) or the lymph nodes. The incomplete endothelial lining due to rapid angiogenesis at the tumor site results in the formation of large and irregular pores (0.1–3 µm) [32], through which nanoparticles can escape from the circulatory stream and accumulate at the TME via a so-called enhanced permeability and retention (EPR) effect [32]. Toy et al. demonstrated that, within the size range of 60–130 nm, smaller particles exhibited greater lateral drift towards the blood vessel walls, which is a prerequisite for interaction with the tumor vascular bed and essential for escape into the TME [33]. Furthermore, both blood clearance and uptake by the mononuclear phagocyte system (MPS) at the liver and spleen depend on the size and composition of PLGA carrier systems. The slit size in the inter-endothelial cells of the spleen is about 200 nm, which facilitates the leakage and circulation of smaller particles for longer time periods [34] or enables their entry to the lymphatic system through direct drainage to the lymph nodes [35]. On the other hand, larger particles, which are not eligible for intravenous administration due to their excessive size, are more likely to be taken up by immune cells at the injection site, such as in the case of subcutaneous or intramuscular pathways [36].

Surface functionality also influences drug pharmacokinetics in vivo. The use of polyethylene glycol (PEG) has been shown to significantly increase circulation time in several studies [36,37,38]. When attached to nanoparticles’ surface (a process so-called PEGylation), the hydrophilicity of PEG chains recruits specific proteins from plasma, that cloak and limit the interactions of particles with MPS cells, hence prolonging blood circulation (‘stealth effect’) [39,40]. The decrease in the aggregation, opsonization, and phagocytosis of particles entails the extension of their circulation time. Morikawa et al. reported additional benefits of PEGylation such as reduced particle size and improved encapsulation efficiency for curcumin-loaded nanoparticles [41]. 

In addition to particle size and surface functionality that have a large influence on the therapeutic efficacy in vivo, particle uniformity in terms of size and encapsulated cargo is also an important aspect particularly for the clinical translation of polymeric drug delivery formulations [42,43]. The uniformity of particles depends largely on the utilized manufacturing approach. The conventional production techniques based on emulsion solvent diffusion, emulsion solvent evaporation, and nanoprecipitation are suitable for the production of sub-micron and micron-size PLGA particles [7,8]; however, these methods lack the precision and full control over the particle size and uniformity particularly for larger scale processes [8]. In recent years, microfluidics technology has emerged as an effective tool to produce particles in a highly controllable manner [44,45]. Major advantages of microfluidics-based particle manufacturing include the requirement of low sample volumes, high surface area, and reduced system footprint [46]. This technology allows for rapid fluid mixing at the nanoliter scale and production of PLGA particles with highly specific sizes as well as surface functionalities only by altering specific parameters such as concentration of the starting materials or fluid flow rates through micron-size channels [41,47,48,49,50,51,52]. Additionally, traditional small-scale laboratory synthesis techniques suffer from batch-to-batch variations while microfluidics technology provides a precise size control, a high degree of particle uniformity, and reproducibility. Furthermore, formulations can be scaled up by increasing the quantities of fluids pumped through the system or by parallelizing multiple microfluidic mixers [53].

In this study, we report on the use of microfluidics technology as a platform for the production of PEGylated PLGA particles. We explored the feasibility to generate different sizes of particles with low polydispersity index (PDI) by varying process and formulation parameters such as total flow rate and flow rate ratio of organic and aqueous phases as well as surfactant and polymer concentration. We tuned the particle size at sub-micron (~100 nm and ~200 nm) and micrometer (>1000 nm) length scales, which represent biologically-relevant cut-off values that influence the particle biodistribution and clearance in vivo. A fluorescent dye was used as a model drug to assess encapsulation efficiency, release profile, and uptake by mouse-derived immune cells. Clearance of systemically administered, fluorescently labeled particles was also studied on a mouse model through in vivo imaging.

## 2. Experimental

### 2.1. Materials

PLGA (Resomer RG 502H), with a 50:50 ratio of lactic acid:glycolic acid and Mw 7000–17,000 Da was obtained from Evonik Nutrition and Care GmbH (Darmstadt, Germany). PEG-PLGA copolymer (PEG M_n_ 5000, PLGA M_n_ 7000), polyvinyl alcohol (PVA, 9000–10,000 Mw, 80%, hydrolyzed) and cholesteryl BODIPY™ FL C_12_ were obtained from Thermo Fisher Scientific (Waltham, MA, USA). Near-infrared emitting fluorescent dye VivoTag-S 750 was purchased from Perkin Elmer Inc. (Waltham, MA, USA) and acetonitrile (ACN, 99.95%) was from VWR International (Radnor, PA, USA). Ultrapure Milli-Q^®^ water (18.2 MΩ.cm) was used where necessary (Merck KGaA, Darmstadt, Germany). Roswell Park Memorial Institute (RPMI) 1640 medium, Anti-Anti (AA), and ß-mercaptoethanol were obtained from Gibco (Thermo Fisher Scientific, Waltham, MA, USA). Granulocyte-macrophage colony-stimulating factor (GM-CSF) was obtained from Peprotech Inc. (Rocky Hill, NJ, USA). X-Vivo medium and ultraglutamine were from Lonza Group (Basel, Switzerland). Fetal bovine serum (FBS) was purchased from Hyclone Laboratories Inc. (GE Healthcare, Chicago, IL, USA).

### 2.2. Equipment

The microfluidics system was set up by connecting syringe pumps (Harvard PHD-2000 infusion 70–200) to a Y-junction mixer with staggered herringbone ridges (NanoAssemblr™, Precision Nanosystems Inc., Vancouver, Canada) through 0.8 mm polytetrafluoroethylene (PFTE) tubing (ID 0.8 mm, OD 1.58 mm) obtained from Sigma-Aldrich (St. Louis, MO, USA). A detailed characterization of the mixing geometry was reported in [44]. Two 20 mL NORM-JECT Luer Lock syringes were connected to 0.8 mm diameter needles (Braun Sterican 0.8 × 120 mm), which were inserted in the fittings connected to the inlets of the mixing cartridge. A PFTE tubing connected to the chip outlet was used for sample collection.

### 2.3. Preparation of PLGA Particles

Prior to particle production, the pipes and the mixing chip were primed first with the solvents (ACN for the organic phase inlet and MilliQ for the aqueous phase inlet) and then with the appropriate phases for 1 min at organic: aqueous flow rates of 2:2. After the priming step, pumps were operated at the desired flow rates. The product obtained within the first 2 min was discarded. The collected particles were left stirring overnight (350 rpm) at room temperature for organic solvent evaporation. A schematic representation of the process is shown in Scheme 1.

The particle size was tuned via varying the flow rates of organic and aqueous phases as well as PLGA and PVA concentrations. Upon determination of the optimized parameters for each target size, PEGylated PLGA particles encapsulating fluorescent dyes emitting either at visible or near-infrared regions were prepared by the following methods: 

>1000 nm particles: the organic phase (5 mL ACN) contained 233.1 mg PLGA, 99.9 mg PEG-PLGA, and 50 μL of 1 mg/mL fluorescent dye in ethanol. A 3% PVA solution was used as the aqueous phase. Particles were produced at 6:2 organic:aqueous flow rates in triplicate. 

~200 nm particles: the organic phase (5 mL ACN) contained 116.6 mg PLGA, 50 mg PEG-PLGA, and 50 μL of 1 mg/mL fluorescent dye in ethanol. A 1% PVA solution was used as the aqueous phase. Particles were produced at 2:6 organic:aqueous flow rates in triplicate.

~100 nm particles: the organic phase (8 mL ACN) contained 93.5 mg PLGA, 40 mg PEG-PLGA, and 80 μL of 1 mg/mL fluorescent dye in ethanol. A 1% PVA solution was used as the aqueous phase. Particles were produced at 4:6 organic:aqueous flow rates in triplicate.

After evaporation of the organic solvent, particles were washed three times with MilliQ water by centrifugation at 15,000 rpm for 35 min (for >1000 nm and ~200 nm particles) and by spin filtration using a 100,000 kDa MW cut-off centrifugal filter device (Millipore, Merck KGaA, Darmstadt, Germany) for ~100 nm particles. Particles were then lyophilized. 

### 2.4. Colloidal Characterization of PLGA Particles

In total, 1.25 mg of lyophilized particles were dispersed in 1 mL of MilliQ water to determine the size distribution and polydispersity index (PDI) with dynamic light scattering using a Nanotrac Flex (Microtrac GmbH, Krefeld, Germany). Zeta potential measurements were obtained with a Zetasizer Nano ZS (Malvern Instruments, Malvern, United Kingdom). Prior to measurements, particles were suspended in a 50 mM NaCl solution. The average of three measurements were used to report the size, PDI, and Zeta potential for each sample.

Atomic force microscopy (AFM) images of particles were obtained with a Catalyst BioScope (Bruker, Billerica, MA, USA) coupled to a confocal microscope (TCS SP5II, Leica Mycrosystems, Wetzlar, Germany). Then, 50 μL of 10 mg/mL particle suspension was dried on clean glass substrates and particles were imaged in peak-force tapping mode using a silicon nitride cantilever with a nominal spring constant of 40 N/m (Bruker). AFM images were analyzed using NanoScope analysis software (Bruker, Billerica, MA, USA).

### 2.5. Determination of Encapsulation Efficiency

Dye loading was quantified by construction of a calibration curve using a BODIPY-C_12_ standard. A series of dye concentrations (in 0–0.25 mg/mL range) were prepared in a solvent composed of equal parts of ACN and MilliQ and fluorescence intensities were measured for each standard. The fit function applied to the linear portion of the curve was used for the calculation of the dye content of particles. The encapsulation efficiency was determined by comparing the total amount of dye in the lyophilized particles to the initial amount of dye supplied for the collected volume.

### 2.6. In Situ Release Profile

PLGA particles containing BODIPY-C_12_ were suspended in PBS in 5 mg/mL concentrations and were dialyzed at 37 °C for 14 days using dialysis tubes with a 1000 Da molecular weight cut-off membrane (GE Healthcare, Chicago, IL, USA). At different incubation times, the dialysis medium was collected for fluorescence measurements using an LS 55 Perkin Elmer fluorescence spectrometer (Waltham, MA, USA). Samples were excited at 488 nm and emission was recorded between 500 nm and 700 nm. After each measurement the dialysis medium was refreshed. Samples were studied in triplicate. 

## 3. In Vitro Cellular Uptake Experiments

### 3.1. Generation of Bone Marrow-Derived Dendritic Cells (BMDCs)

Dendritic cells were generated from mouse bone marrow cells by culturing them in full RPMI medium containing 5 mL 200 mM l-glutamine (Gibco, Thermo Fisher Scientific, Waltham, MA, USA), 5 mL 100× antibiotics and antimycotics (Gibco, Thermo Fisher Scientific, Waltham, MA, USA), 10% fetal bovine serum (Gibco, Thermo Fisher Scientific, Waltham, MA, USA), and 500 μL β-mercapto ethanol (Sigma-Aldrich, St. Louis, MO, USA). Then, 5.0 × 10^6^ cells in 13 mL of full medium were cultured in the presence of 20 ng/mL GM-CSF for 7 days. At day 3, 4 mL of complete medium containing 37.9 ng/mL GM-CSF was added. At day 6, 1 mL complete medium containing 158 ng/mL GM-CSF was added. Cells were used for uptake experiments at day 7.

### 3.2. Isolation of Myeloid-Derived Suppressor Cells (mMDSCs) and Polymorphonuclear Myeloid-Derived Suppressor Cells (pmnMDSCs)

Gr-1^dim^Ly-6G^−^ monocytic and Gr-1^high^Ly-6G^+^ polymorphonuclear myeloid-derived suppressor cells (mMDSCs and pmnMDSCs, respectively) were isolated from the spleen of tumor-bearing mice using an isolation kit (Miltenyi Biotec, Bergisch Gladbach, Germany) according to manufacturer’s instructions. Briefly, the spleen was isolated under sterile conditions and meshed through a 100 μm cell strainer with a syringe plunger. The cell suspension was spun at 400× *g* for 5 min and resuspended in 3 mL of 1× ammonium chloride solution for the lysis of erythrocytes. After 5 min of incubation at room temperature, cells were washed with 10 mL of PBS. The cells were incubated with an Anti-Ly-6G-Biotin antibody and Anti-Biotin MicroBeads and were subsequently applied to a magnetic-activated cell sorting (MACS) column, which retained the pmnMDSCs. The flow-through containing mMDSCs were eluted as the positively selected cell fraction and were further purified by applying them to a second MACS column.

### 3.3. In Vitro Cellular Uptake

Firstly, 1.0 × 10^5^ cells in 500 μL complete medium were transferred to 5 mL propylene round bottom tubes (Falcon). Then, 10 μg of particles containing BODIPY-C_12_ water were added to the round bottom tubes and were incubated for time periods of 1, 2, 4, 6, 24, and 48 h. After incubation, particle uptake was determined by flow cytometry analysis on a FACSVerse (BD Biosciences, Franklin Lakes, NJ, United Sates). 

## 4. In Vivo Clearance Studies

All animal experiments were performed according to guidelines of Radboud University’s Animal Experiment Committee and Central Authority for Scientific Procedures on Animals (project number 2015-019TIL, date September 2015) in accordance with the ethical standards described in the Declaration of Helsinki. Wild-type BALB/cAnNCrl mice, aged 8–12 weeks, were obtained from Charles River, Germany and maintained under specific pathogen-free conditions at the Central Animal Laboratory (Nijmegen, The Netherlands). Drinking water and food were provided ad libitum. Mice were warmed either in a heating chamber or under a heating lamp and 1 mg PLGA nanoparticles (~200 nm and ~100 nm) containing VivoTag-S 750 were injected in 200 µL of phosphate-buffered saline (PBS) solution through a lateral tail vein using a 1 mL syringe with a 29 G needle. Then, 0.5, 3, 24, and 48 h after injections mice were shaved and imaged in an IVIS Lumina II (Perkin Elmer) system. Mice were euthanized, and organs were dissected and imaged separately at 24 and 48 h. Imaging settings were: exposure time: 3 s; binning: medium; F/stop: 2; fluorescent excitation filter: 745 nm; fluorescent emission filter: 810–885 nm. A fluorescent background acquisition was performed for each time point. Living Image software (Caliper Life Sciences, Hopkinton, MA, USA) was used for data analysis. Background values were subtracted from measurement values. Same sized regions of interest (ROI) were applied on the liver and bladder for full body image analysis; also, same sized ROIs were applied on the isolated liver and spleen. Total flux (photon/s) per each ROI was calculated. 

### Statistical Analysis

An unpaired *t*-test was used to determine the significance of the difference in mean particle size values among the compared groups using GraphPad Prism.

## 5. Results and Discussion 

### 5.1. Preparation of Particles

A commercially available NanoAssemblr™ cartridge [8] connected to syringe pumps was used for the production of PLGA particles through nanoprecipitation. In this method, water-miscible organic solvents are used to dissolve the polymer which, once at the interface between the aqueous medium and the organic solvent, will start to precipitate. Polymer deposition caused by fast diffusion of the solvent leads to instantaneous formation of a colloidal suspension and particle size is mainly determined by solvent diffusion coefficient (*D*) and mixing time (*t*_mix_) [54]. The architecture of the mixing chip also plays a crucial for the generation of a hydrodynamic-focusing flow pattern and the mixing process can be controlled by varying the width of the focused stream (*wf*) as shown in Equation (1) [44]:(1)$$t_{mix}\approx \frac{wf^{2}}{4D}$$

By varying the ratio of flow rates, the lateral width of the focused stream can be adjusted very accurately, which leads to a highly controlled diffusion and mixing at the interface of organic and aqueous phases. In our study, acetonitrile (ACN) was used as the organic phase and particles were formed upon rapid mixing of organic and aqueous phases in micron-size channels. Due to rapid diffusion of ACN to the aqueous phase, PLGA precipitated and was subsequently stabilized by PVA present in the aqueous phase. Process parameters such as (total) flow rates of organic and aqueous phases as well as formulation parameters such as PLGA and PVA concentration were varied to tune the size of PLGA particles from sub-micron to micron-scales.

The first process parameter tested was the flow rate of the organic phase composed of a 33.3 mg/mL PLGA solution in ACN. The organic phase flow rate was varied between 2 mL/min and 6 mL/min while the aqueous flow rate was kept constant at 2 mL/min for a 1% PVA solution (Figure 1A). A gradual increase in the particle size from ~400 nm to ~900 nm was observed with an increase of the organic phase flow rate. This was most likely caused by precipitation of higher amounts of PLGA at a given mixing time when relatively higher organic phase flow rates were used. Similarly, increasing the flow rate of the aqueous phase for a given set of formulation parameters and organic phase flow rate resulted in the formation of smaller particles (Figure 1B). Doubling the aqueous flow rate to 4 mL/min resulted in particles almost half the size of those produced at 2 mL/min. However, increasing the aqueous flow rate up to 6 mL/min did not result in a notable change in particle size. 

Another process parameter affecting particle size was the total flow rate of the organic and aqueous phases. With the equal flow rate of organic and aqueous phases, the increase of the total flow rates from 4 mL/min (2:2) to 8 mL/min (4:4) led to a decrease in the particle size (Figure 1C). Increasing the total flow rate further to 12 mL/min (6:6), however, did not cause any notable difference in the particle size other than a slight improvement in the PDI. Increasing the total flow rates from 8 mL/min to 12 mL/min apparently did not cause a relevant variation of solvent diffusion time to induce a significant change in particle size. The total flow rate determines both the mixing time of the two phases in the cartridge and the collection rate of the particles at the outlet channel. Faster mixing by increasing the total flow rates has also been reported in other studies to result in the formation of smaller particles [41,47]. 

Overall, particles in the sub-micron size range were obtained by varying only the flow parameters. Furthermore, we varied the PLGA and PVA concentrations in order to increase the particle size to micron scale. A representative data set in Figure 1D shows that particle size almost tripled when the PLGA concentration was doubled. In these batches, the PVA concentration was increased as well and organic:aqueous flow rates were kept equal (4:4 mL/min). The larger particle size obtained with a higher PLGA concentration can be explained by the higher viscosity of the organic phase, which resulted in a slower diffusion of ACN to the aqueous phase and an increased mixing time [55]. 

For all tested conditions, a good batch-to-batch reproducibility and a small PDI (≤0.2) was observed. After several trials, the optimal conditions and parameters were determined to obtain particles of sizes >1000 nm, ~200 nm, and ~100 nm. These parameters were then used to prepare particles with a surface functionalized with polyethylene glycol (PEG) and encapsulating a fluorescent dye (Table 1). 

For fluorescent labeling, two different fluorescent dyes with visible (BODIPY™ FL C_12_, green, 510 nm) and near-infrared (VivoTag-S 750, red, 750 nm) emission wavelengths were used. BODIPYs are low-polarity dyes with stable fluorescence emission properties [56] and are commonly used in fluorescence detection and photodynamic therapy applications [57]. VivoTag-S 750 is a near-infrared emitting (NIR) fluorochrome extensively used for in vivo imaging applications [58,59]. PLGA particles labeled with the green fluorescent dye were used for in situ release and in vitro uptake experiments, whereas those labeled with the red dye were used for in vivo imaging studies. 

### 5.2. Colloidal Characterization

A detailed colloidal and functional characterization of PEGylated PLGA particles encapsulating BODIPY-C_12_ is shown in Figure 2. Both atomic force microscopy (AFM) images (Figure 2A–C) and dynamic light scattering (DLS) measurements (Figure 2D) revealed the formation of monodisperse particles (PDI ≤0.2) within the desired size range (i.e.; >1000 nm, ~200 nm, ~100 nm) also upon PEGylation and fluorescent dye encapsulation. It should be noted that although PDI values smaller than 0.3 are considered acceptable for drug delivery applications, more specific standards and guidelines have yet to be established by regulatory authorities. 

The micron-sized particles showed an average size of 1690 nm (±60 nm), while the sub-micron particles were 250 nm (±48 nm) and 106 nm (±5 nm) in diameter. For ease of reporting, we will continue referring to them as >1000 nm, ~200 nm, and ~100 nm particles. Slightly negative ζ potential values that spanned a range between −5 mV and −15 mV were observed for all particle types (Figure 2E). While the variation of ζ potential was negligible among sub-micron particles (i.e., ~200 nm, ~100 nm), the micron-size particles displayed the most negative value. PLGA nanoparticles stabilized with PVA have also been reported to bear a slightly negative ζ potential (−5 mV) [60] due to the presence of residual PVA on the particle surface, which affects the number of carboxylic acid end groups [61]. The hydrophobic acetate moieties in partially hydrolyzed PVA can lead to its entrapment within the PLGA matrix on the particle surface, thereby masking the surface charges to almost neutral ζ potential values [62]. Therefore, the more negative ζ potential obtained for micron-size particles can be related to differences in the PVA content and the flow rate of the aqueous phase for micron-size particles compared to sub-micron particles, as well as higher PLGA content. Apparently, these variations in formulation and process parameters influence not only the particle size but also the particles’ surface properties, which collectively determine the pharmacokinetic properties, cellular uptake, and particle biodistribution [63].

Encapsulation efficiency of BODIPY-C_12_ was determined using fluorescence spectroscopy by comparing the initial amount of dye supplied during particle production and the total amount of dye detected in the entire yield. The efficiency of dye encapsulation was significantly higher for smaller particles with the highest efficiency at 44% for ~100 nm particles, which decreased gradually to 21% and 13% for ~200 nm and >1000 nm particles, respectively. It should be noted that each particle type was produced using different formulation and process parameters, which equally play a role in determining the encapsulation efficiency [64]. An efficient encapsulation requires the rapid precipitation of polymer with the fluorescent dye to restrict the dye within the polymer matrix and prevent its diffusion to the aqueous phase. The micron-size particles with the lowest encapsulation efficiency were prepared using the highest polymer concentration and the highest organic:aqueous phase flow rate ratios. The high viscosity of the organic phase as well as the relatively low fraction of the aqueous phase probably resulted in a slower polymer precipitation, and therefore a lower encapsulation efficiency for micron-size particles. 

The functional characterization of particles was achieved by monitoring the dye release. For this study, particles dispersed in PBS were dialyzed at 37 °C for a period of 14 days. At certain time points, fluorescence intensities of the dialysis media were measured using fluorescence spectroscopy. The dialysis medium was replaced with a fresh medium after each measurement. Each formulation type showed a distinct release profile that was strongly correlated with the particle size. After an initial burst, the release was steady for all particles (Figure 2F). The magnitude of burst release was higher for ~100 nm particles (approx. 30%), which was found as ~15% and ~10% for ~200 nm and >1000 nm particles, respectively.

In addition to the highest burst release, the overall release rate was also faster for ~100 nm particles such that they already released the majority of their content (~80%) by day 4. On the other hand, the released content of ~200 nm and >1000 nm particles barely reached 50% and 20%, respectively, by the end of the entire release period (two weeks). Indeed, several different release patterns such as mono-, bi-, and tri-phasic release are reported for PLGA particles, which are mainly regulated by the physicochemical properties of the cargo, PLGA type (molecular weight, lactide/glycolide ratio, etc.), and particle morphology (size, porosity, etc.) [65]. Since the same cargo and polymer type were used for the preparation of particles, the main reason for different release rates observed was the particle size. Larger surface area/volume ratio of smaller particles facilitated the faster diffusion of dye molecules located on or close to the surface. Such inverse relation between the release rate and particle size is in good agreement with the general trends reported in other studies [66,67,68].

### 5.3. In Vitro Uptake Experiments

Particle size and surface functionality are among the key parameters that influence their interactions with cells [69]. In this work, we used mouse-derived cells, namely bone marrow-derived dendritic cells (BMDCs), CD103^+^ dendritic cells (CD103^+^), and myeloid-derived suppressor cells (MDSCs) of monocytic (mMDSCs) and polymorphonuclear (pmnMDSCs) sub-types to study the uptake of fluorescently labeled, PEGylated PLGA particles of varying sizes (Figure 3).

Particles incubated with BMDCs for different time periods were analyzed using flow cytometry (Figure 3A). In our previous study we reported that, once taken up by the cells, the integrity of PLGA particles is compromised before 72 h of incubation [70]. Hence, in the present work, we monitored particle uptake up to 48 h in order to avoid possible variations in the mean fluorescence intensities (MFIs) due to particle degradation at later time points. For ease of comparison, MFI data were normalized to 1 for the values obtained at 48 h of incubation time. A clear correlation between the particle size and the trend in particle uptake was observed. For sub-micron particles (~100 nm and ~200 nm) a stable MFI was observed within the first 6 h, which increased at later time points (Figure 3A, red and blue curves). On the other hand, micron-size particles (>1000 nm) showed a time-dependent uptake behavior. For these particles, the intracellular MFI increased gradually within the first 6 h, reaching a plateau until the end of the incubation period (Figure 3A, orange curve). 

Overall, the uptake of sub-micron particles was more efficient compared to micron-size particles. The MFI obtained at 1 h of incubation already corresponded to ~75% and ~50% of the total MFI for 100 nm particles (Figure 3A, red) and 200 nm particles (Figure 3A, blue), respectively. It should be noted that each type of PLGA particles had different dye loading ratios. In this uptake study, equal amounts of PLGA particles with different total fluorescence intensities were used. Although normalization of MFI enabled a fair comparison of the particle uptake trend for different sizes, we further investigated the particle uptake on other types of mouse-derived immune cells using particles with equal fluorescence intensities instead of equal particle mass in order to compensate for the variations in dye encapsulation (Figure 3B–D). Flow cytometry histograms of the data presented in Figure 3B–D are shown in Appendix A. A clear correlation between the particle size and intracellular MFI was observed for the cells that were either generated in vitro (Figure 3B) or were isolated from the spleen of tumor-bearing mice (Figure 3C,D). After 2 h incubation period, the uptake of ~100 nm particles was ~1.5 fold higher than the >1000 nm particles for CD103^+^ dendritic cells (Figure 3B) and mMDSCs (Figure 3C), and the difference was even higher (almost three-fold) for pmnMDSCs (Figure 3D). Similar studies also reported the lower uptake efficiencies of micron-size particles compared to sub-micron particles by dendritic cells [71], which could be due to different uptake mechanisms associated with different particle sizes. The uptake of small particles (<100 nm) has been reported to be clathrin- and/or caveolin-mediated [72], whereas micropinocytosis and phagocytosis are the main mechanisms via which sub-micron particles (~200 nm) and micron-sized particles (>1000 nm) are taken up [73]. In addition to particle size, surface charge is an important parameter that influences the efficiency of particle uptake by cells. Prior to their internalization, particles need to attach on the cell surfaces that are decorated with negatively-charged proteoglycans [63]. Consequently, the uptake of positively-charged particles can be more efficient due to electrostatic interactions between the particles and cell surface. In our work, all particles had negative ζ potential values, which was highest for >1000 nm particles. Therefore, the poorer uptake efficiency observed for micron-size particles can be due to less-favorable surface charge in addition to larger particle size.

All the cell types used in our work are important regulators of immune response. BMDCs and CD103^+^ are antigen presenting cells that can prime T cells to induce antigen-specific immune responses [74]. On the other hand, MDSCs play an important role in immune suppression in cancer as well as in tumor angiogenesis, drug resistance, and promotion of tumor metastases [75], representing an attractive potential therapeutic target, for things such as cancer immunotherapy. Among the studied PLGA formulations, ~100 nm particles with almost neutral surface charge would be the most efficient vehicle to deliver such things as antigens to dendritic cells (DCs) and immunomodulatory drugs to MDSCs for cancer immunotherapy. 

### 5.4. In Vivo Clearance of Particles

Rapid clearance of particles from the bloodstream through the mononuclear phagocyte system (MPS) and reticuloendothelial system (RES) represents a major limitation to achieve preferential accumulation of particles in the target organs [76]. Tuning the size and surface properties of particles has proven useful in preventing their rapid removal from the bloodstream [77,78]. In order to evaluate particle clearance in vivo, we administered PLGA particles labeled with a near-infrared (NIR)-emitting dye (VivoTag-S 750) intravenously (i.v.) to mice. The acceptable particle size range for the i.v. injections have been reported as 10–1000 nm to prevent possible accumulation of larger particles in the lung capillaries [61,79,80]. Therefore, we used only sub-micron size particles for in vivo studies. The colloidal properties of ~100 nm and ~200 nm particles encapsulating the NIR-emitting dye were similar to those labeled with the green fluorescent dye (Appendix A). Whole-body and ex-vivo organ imaging were performed at different time points up to 48 h after i.v. administration of sub-micron PLGA particles (Figure 4).

Whole-body imaging revealed the presence of both particle types mainly in the liver and bladder already after 30 min following the injection (Figure 4A). The liver signal decreased gradually at later time points and was still above background levels at 48 h for both ~100 nm and ~200 nm particles. The presence of a high fluorescence signal in the bladder at 0.5 h was an interesting observation, which could be related to the excretion of free dye molecules that were burst-released. In fact, intact particles cannot pass through the renal filtration barrier since it has an effective size cut-off of ~10 nm [81]. The higher percentage of burst release displayed by ~100 nm particles in situ aligned well with the in vivo observations, in which the bladder signal resulting from the excretion of the free dye molecules was relatively lower for ~200 nm particles. Overall, the bladder signal did not provide a reliable measure of the systemic clearance of the particles due to the interference of the burst-released dye. Thus, we monitored the clearance of particles from the liver as representative of their systemic clearance. The influence of burst-released dye can be avoided via covalent attachment of the fluorescent dye. 

The variation of liver signal was monitored using the whole-body images obtained at different time points (Figure 4B). For a better comparison, the values were normalized to maximum intensities measured at 3 h. From this point on, the decay of liver signal was almost linear for ~100 nm particles with a slope of −0.014 (R^2^ = 0.9926) (Figure 4B, blue curve), whereas the decay of ~200 nm particles better suited an exponential function (Figure 4B, green curve). Since the number of fit points was not sufficient for an accurate exponential fit, we used a linear fit for ~200 nm particles as well. These particles displayed a relatively faster decay with a slope of −0.018 (R^2^ = 0.9537). Ex-vivo imaging of an isolated liver also showed similar variations in signal intensities, such that ~200 nm particles (Figure 4C, green curve) displayed a more pronounced decrease at 48 h compared to ~100 nm particles (Figure 4C, blue curve). Faster clearance of larger particles was also shown in other studies. When administered intravenously, particles larger than 200 nm activate human complement systems, are rapidly eliminated from the bloodstream, and gather mainly in the liver and spleen where the rate of accumulation is proportional to the size of the particles [30]. While particles with a diameter greater than 200 nm were most likely cleared by Kupffer cells, smaller particles displayed a decreased rate of clearance and an extended circulation time [82]. Of note, we did not observe a signal originating from the spleen in whole-body imaging. However, ex-vivo imaging of isolated spleen at 24 h and 48 h revealed signal intensities that were only slightly above the background noise, indicating that the spleen was not the preferential accumulation site for sub-micron PLGA particles (Appendix A). Indeed, it has been reported that the spleen receives barely 15% of the i.v. injected dose of nanomedicines [83]. Therefore, additional surface modification strategies may be needed for applications that require splenic accumulation of PLGA particles. 

## 6. Conclusions

In this study, we demonstrated how PLGA particle size can be specifically tuned using a microfluidics system via modulating the formulation and process parameters. Through a series of optimization experiments, we obtained PEGylated PLGA particles in different sizes, which remarkably affected the characteristics of the particles in vitro and in vivo confirming the direct relation between the size and the pharmacokinetics behavior. This work can be considered as a further step towards the establishment of a production process that is able to generate tailor-made medicine for each individual clinical need.

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
