# Peer review of "Microfluidics-Assisted Size Tuning and Biological Evaluation of PLGA Particles"

_pharmaceutics, 2019, doi:10.3390/pharmaceutics11110590_

Round 1

Reviewer 1 Report

The Manuscript presents Microfluidics assisted preparation of PLGA nanoparticles (NP) by nanoprecipitation mechanism. Authors evaluated effect of process and formulation parameters on size of NP. Functional characterization and biological evaluation (in vitro uptake and tissue distribution) were presented. Experimental design is adequate and results are explained in detail. Overall manuscript is well written.

Comments:

Explain why particle size did not change with increase in total flow rate to 12 mL from 8mL, while there was a significant change between 4 mL and 8 mL. Not clear why flow rates of 2:6 and 4:6 were selected to prepare particles of ~200 nm and 100 nm size, respectively. Revise Figure 1 legend to correct errors. Define abbreviations as they first appear in the manuscript.

Reviewer 2 Report

The present paper presents the application of microfluids based technology method to produce the biodegradable PLGA particles for application as drug carriers. The developed method enables the reproducible production of the particles of the desired sized. The particles of various size  (1600, 250 and 100 nm) were produced and completely characterized using various physico chemical methods. Efficiency of particle uptake by was tested by dendritic cells and myeloid-derived suppressor cells isolated from mice. The experiments were carefully conducted. All experimental results were well exxplained and represent the new approach in using nanomaterials in medicine. I have no other comments and recommend to publish the paper as it is.  

Author Response

We thank Reviewer 2 for their time to review our manuscript and for their comments.

Reviewer 3 Report

In this study nano- and microparticles composed of PLGA and PEG-PLGA copolymers were prepared using microfluidics technology. This technology for the mentioned polymers is well-known (e.g. reviewed in S. Rezvantalab, M.K. Moraveji: Microfluidic assisted synthesis of PLGA drug delivery systems. (2019) RSC ADVANCES ,9 (4), 2055-2072.). The main novelty of the present study might be the in vivo analysis, however, it contains a substantial mistake. Overall, I do not evaluate this work strong enough for the publication in the journal ‘Pharmaceutics’ for the following reasons:

Major comments:

As authors state one of the main advantage of this technique is achievable monodisperse size. In contrast, according to the AFM images, the monodispersity of the particles with about 1 micrometer is questionable. Furthermore, nanoparticles with size ~100 nm and ~200 nm had size distribution with PDI higher than 0.1 which does not suggest really narrow size distribution. In my PLGA preparation experiences both with emulsion and nanoprecipitation techniques, PDI>0.1 is not satisfactory. In page 10, first paragraph authors write that ‘After 2 h incubation period, the uptake of ~100 nm particles was ~1.5 fold higher than the >1000 nm particles for CD103+ dendritic cells (Figure 3B) and mMDSCs (Figure 3C), and the difference was even higher (almost 3 fold) for pmnMDSCs (Figure 3D).’ Since the fluorescent dye encapsulation efficiency (thus also the loading) of particles with size ~100 nm was 44 %, while 13 % for >1000 nm particles, this comparison fails. For a more comprehensive comparison authors should represent the FACS results that show the ratio of cells, which contain fluorescently-labelled NPs (if it shows significant differences for the particles with different size). The burst release of fluorescent dye made the in vivo study uncertain. Seeing the release results the fluorescent dye should have been attached chemically to the nanoparticles in order to achieve a correct in vivo test.

Minor comment:

Figs. 1 and 2. title of Y axis ont he left should be Size (nm) or a used term, MI is not defined and understood.

Reviewer 4 Report

The manuscript is well-written and presented. The studies address a common problem in nanoparticle production - that of reproducible production of polymeric and lipid-based particles. The authors have systematically examined typical parameters related to microfluidic production of PEG-PLGA particles with encapsulation of a dye. They further examine release of the dye in vitro and track the biodistribution of the particles in vivo.

The strongest part of the manuscript is the systematic way in which the authors examine the parameters for microfluidic production of PLGA particles. This section is likely to be the most useful section of the paper for the reader as it outlines a basic framework for using microfluidics.

However, this part could be strengthened with an in more depth discussion of how/why the changes in parameters have the effect that they do. There is substantial information (theoretical and experimental) on the mixing effects in microfluidics, and it would have been instructive had the authors tried to explain their results in those terms.

The in vivo studies are not particularly instructive other than showing biodistribution and release of the encapsulated dye - but limited to the liver and bladder. The imaging  studies should have been backed up by more in-depth ex-vivo examination of particles in the different organs including blood.

In the release studies, the authors say they have normalized the results - what does this mean exactly? Please show how the calculations were done to normalize the fluorescence results. Where differences are seen, an attempt to calculate the significance or non-significance of the results should also be done. 

Round 2

Reviewer 3 Report

The questions has been responded, and necessary corrections have been made.